# Observation of quantum criticality of a four-dimensional phase transition

**Farid Madani** ⓘ **, Maxime Denis** ⓘ **, Pascal Szriftgiser** ⓘ **, Jean-Claude Garreau** ⓘ **, Adam Rançon** ⓘ **& Radu Chicireanu** ⓘ ✉

Understanding how a system's behavior extrapolates beyond 3D is a fundamental question in physics, spanning topics from unification theories to critical phenomena. In statistical physics, fluctuations' strength is highly sensitive to dimensionality, affecting phase transitions. In low dimensions, phase transitions are suppressed, while high-dimensional systems exhibit simpler mean-field behavior. In some cases, like the Anderson localization-delocalization transition in disordered media, criticality remains non-trivial even in dimensions larger than three, presenting challenges to existing frameworks. In this work, using a periodically-driven ultracold atomic gas to engineer disorder and synthetic dimensions, we experimentally observe a phase transition between localized and delocalized phases. The results display three key features of the 4D transition: 1) observables follow d=4 critical scale invariance, 2) critical exponents match numerical predictions for the 4D Anderson transition, and 3) they agree with Wegner's relation in 4D. These findings provide a new avenue for exploring complex critical phenomena in higher dimensions.

A hallmark of criticality near phase transitions, scale invariance stems from the fact that, near the critical point, fluctuations manifest at all length scales. This leads to the concept of universality classes—the property that widely different physical systems can display the same critical behavior, and can thus be described by the same underlying scale-invariant theory. While it may differ dramatically, the behavior within a class will become increasingly similar as the critical point is approached, with the emergence of scaling laws defined by the same critical exponents. In this sense, critical exponents are "universal" and depend only on the symmetries of the system, and on its dimensionality $d$. In many cases, as dimension is increased -and with it the number of neighbors- only the system's average behavior remains relevant, making mean-field theories exact starting from some finite "upper critical dimension" (equal to four in many cases[1–4]). However, a few notable exceptions are known to exist, like the Kardar–Parisi–Zhang equation[5–9]—describing interface growth-and the stochastic Navier-Stokes equation—describing turbulence, for which the upper critical dimension has been argued to be infinite.

Another example is the class of Anderson transitions describing wave transport in disordered media[10], between a localized phase, where subtle destructive interferences suppress completely propagation in the medium, and a delocalized/diffusive phase where transport is possible, see Fig. 1. Previously observed in 3D in a wide panel of physical systems of various natures[11–17], the unconventional nature of the Anderson transitions makes their description in high dimensions an incredibly hard hurdle[18]. There are few and far between theoretical quantitative predictions, one example being Wegner's scaling law[19], relating the exponents on each side of the critical point and the dimension of the system. Yet, no actual theory allows for predicting their precise values, nor computing the scaling functions describing the transition. New approaches are therefore needed to shed light on the unusual critical properties of the Anderson transition, and complex phase transition in general, in high dimensions.

In recent years, various experimental techniques have been devised to create synthetic dimensions by leveraging non-spatial degrees of freedom, and used to control symmetries[20,21], introduce nontrivial topological effects[22–25], or extend the physics where the system is intrinsically limited to lower dimensions[26–28]. In this work, we use a time-modulated driven gas of ultracold atoms to engineer both synthetic dimensions and disorder[29,30], and study a phase transition

Univ. Lille, CNRS, UMR 8523 - PhLAM - Physique des Lasers Atomes et Molécules, F-59000 Lille, France. ✉e-mail: radu.chicireanu@univ-lille.fr

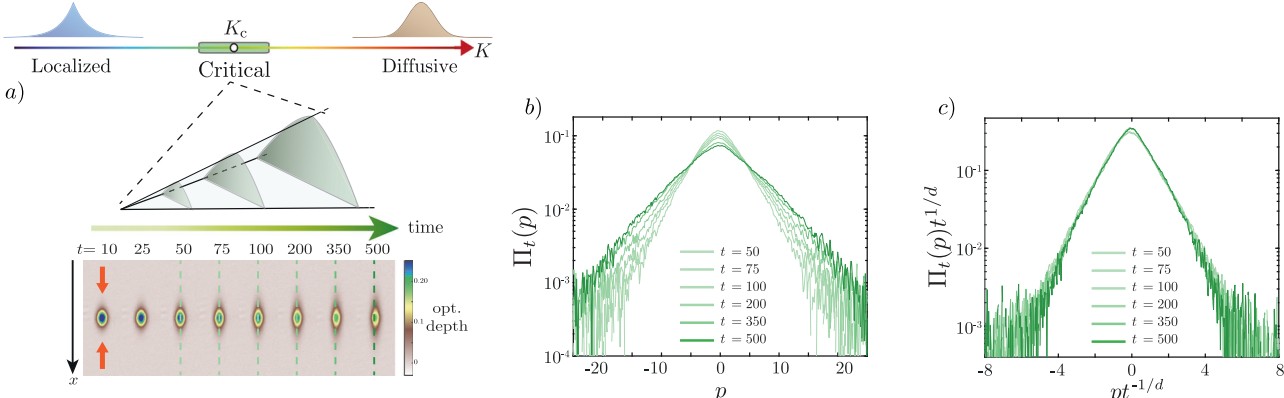

**Fig. 1 | Scale invariance of the 4D Anderson transition. a** Top: Schematical phase diagram of the Anderson transition. As the control parameter $K$ is increased, the system goes through a transition from a localized ($K < K_c$) to a diffusive ($K > K_c$) phase, through a critical point at $K = K_c$. The middle part represents the critical region where the physics is universal. Middle: At the critical point, the physics is scale invariant. As time increases, the observables (here the momentum distribution shown in green) is described by a universal shape that is rescaled as time increases. Bottom: Absorption images of the kicked atomic cloud at different times belonging to the Anderson universality class. The main results of this work are two-fold. The first one is aimed at providing clear-cut experimental proofs of the 4D nature of this phase transition, by (i) observing the specific scale invariance at the critical point, (ii) measuring the critical exponents and (iii) testing the Wegner scaling relation in $d = 4$. The second one is an experimental investigation of the critical scaling in the vicinity of the transition, and a demonstration of the full collapse of the atomic momentum distributions into a single, *two-parameter* scaling function, which provides valuable experimental insights into the complex universal scaling behavior of the Anderson transitions in higher dimensions.

at the critical point, obtained using the 4D QpQKR (1). The color scale represents the optical depth. The sinusoidal potential is applied along the $x$ direction (red arrows). The relevant dynamics is along the $x$ direction, which is decorrelated from the others. **b** Momentum distributions measured in the $x$ direction, in semi-log scale, extracted from the absorption images [shown as dashed lines in **a**)]. Darker colors represent later times. **c** Scale invariance of the momentum distribution at the critical point: after rescaling the distribution and momenta by $t^{1/d}$ with $d = 4$, Eq. (4), all curves collapse onto a single one.

## Results

The system is an atomic kicked rotor[31] experimentally realized using a dilute (non-interacting) cloud of cold atoms submitted to a pulsed, far-detuned 1D optical standing wave (SW). The corresponding Hamiltonian, expressed in dimensionless units, reads

$$H = \frac{p^2}{2} + \mathcal{K}(t)\cos(x)\sum_n \delta(t - n),\qquad(1)$$

where $x$ is the particle position (along the laser axis, measured in units of the inverse of the kick-potential wavenumber $k_K$), $p$ is the conjugate momentum, normalized such that $[x, p] = i\hbar_e$, with $\hbar_e = \hbar k_K^2 T_1/M$ ($M$ is the mass of the atoms, and $T_1$ the kick period) playing the role of a reduced Planck's constant. Time is expressed in units of $T_1$ and $\mathcal{K}$ is proportional to the ratio of SW intensity to its detuning. The SW is produced by a pulsed laser, with a pulse duration of 20 ns, much shorter than the repetition period of ~7 μs, and can be modulated in amplitude to realize in a good approximation the $\delta$-kicked potential in Eq. (1). After applying a given number of pulses, the momentum distribution of the atoms is measured through time-of-flight, see Fig. 1a and Methods section (MS) for further experimental details. If the kick strength $\mathcal{K}(t)$ is constant, the system is effectively 1D and displays dynamical localization, that is, Anderson localization in momentum space[31,32]. Indeed, the kicks change the atoms' momenta, allowing them to "hop" in momentum space, while kinetic energy term gives them a pseudo-random phase in momentum space, playing the role of disorder[33]. By carefully crafting the time-dependence of the $\mathcal{K}(t)$, one can change the system's effective dimensionality[29,30]. This has allowed for a quite thorough investigation of Anderson physics: observation of

the localization in one[31] and two dimensions[34], observation of the 3D transition[14,35], characterization of its critical properties[36,37] and of its universality[38].

For our present purpose, following[14,30,35,39], we realize a quasi-periodic quantum kicked rotor (QpQKR) by taking $\mathcal{K}(t) = K(1 + \varepsilon \cos(\omega_2 t + \varphi_2)\cos(\omega_3 t + \varphi_3)\cos(\omega_4 t + \varphi_4))$, with frequencies $\omega_2/2\pi = \sqrt{5}$, $\omega_3/2\pi = \sqrt{13}$ and $\omega_4/2\pi = \sqrt{19}$ (the angular frequencies $\omega_i$, $\hbar_e$ and $2\pi$ must be incommensurate to avoid resonances, and the universal properties of the transition are independent of these choices as shown thoroughly both experimentally[38] and numerically[40] for the 3D QpQKR). This makes the driving quasiperiodic, but by defining the synthetic dimensions $x_i = \omega_i t + \varphi_i$ ($i = 2, 3, 4$) and their conjugate momenta $p_i$ we can map this Hamiltonian onto a periodic one

$$\mathcal{H} = \frac{p^2}{2} + \omega_2 p_2 + \omega_3 p_3 + \omega_4 p_4 + K(1 + \varepsilon\cos(x_2)\cos(x_3)\cos(x_4))\cos(x)\sum_n \delta(t - n).\qquad(2)$$

Here again, the kicks play the role of a hopping term of the (four-dimensional) momentum while the kinetic energy (both quadratic and linear in momentum) plays the role of disorder. One can show[30,33] that the Floquet eigenstates of the periodic Hamiltonian $\mathcal{H}$ are eigenstates of a 4D disordered tight-binding Hamiltonian in momentum space (MS), thus able to display an Anderson transition with time-reversal symmetry. It turns out that there is no mobility edge, meaning that, for given values of $K$ and $\varepsilon$, all eigenstates are of the same nature, either localized, critical, or delocalized, making a detailed study of the critical properties possible.

### Phase diagram and critical scaling

The phase diagram in the ($K$, $\varepsilon$) space, obtained from numerical simulations (see MS), is shown in Fig. 2a), with a color code corresponding to the behavior of the kinetic energy, proportional to the variance of the momentum distribution $\Pi_t(p)$, $\langle p^2 \rangle_t = \int dp\, p^2 \Pi_t(p)$. For small $K$ and $\varepsilon$ the system is localized (blue region): the momentum distribution freezes at long times with a finite width, $\langle p^2 \rangle_t \to p_{\mathrm{loc}}^2$ with localization length $p_{\mathrm{loc}}$ (in momentum space). In the opposite limit, the system is delocalized/diffusive (brown region): the variance of the momentum distribution increases linearly with time, $\langle p^2 \rangle_t \to Dt$, with diffusion coefficient $D$. The Anderson transition takes place on a critical line $\varepsilon_c(K)$ [red dashed-dotted line in Fig. 2a], and the dynamics is universal and scale invariant in the vicinity of the transition[41], see the

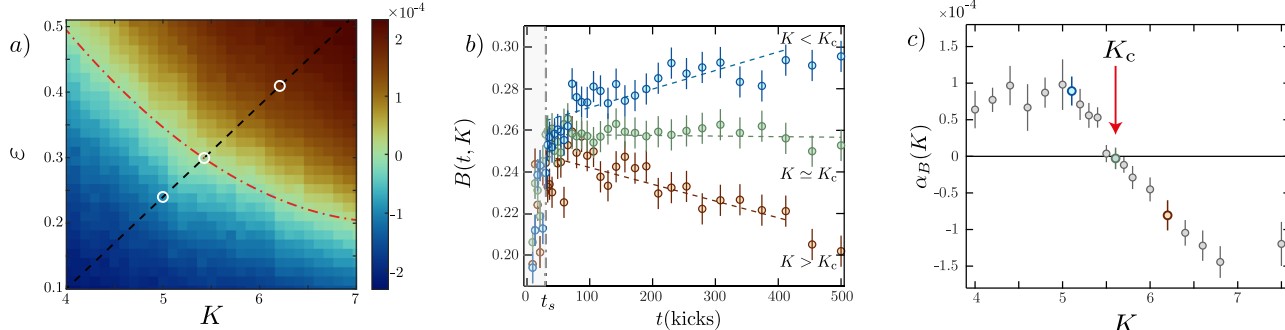

**Fig. 2 | Locating the critical point of the 4D Anderson transition. a** Phase diagram of the QpQKR as a function of $K$ and $\varepsilon$ (numerical simulation). The color scale represents the slope of $\log(t^{2\alpha}\langle p^2\rangle_t)$ vs. $t$, $\alpha = -1/4$, with the blue (brown) region corresponding to the (de)localized phase. The critical line is represented by the red dashed-dotted line along the green region. The dashed black line represents the path in parameter space studied experimentally. **b** Experimental data showing the Binder parameter $B(t,K) = \Pi_t(0)^2\langle p^2\rangle_t$ in the localized ($K < K_c$, blue circles), critical ($K \simeq K_c$, green circles), and diffusive ($K > K_c$, brown circles) regimes (from top to bottom). After a short transient time ($t \lesssim t_s = 30$ kicks, gray area), $B(t,K)$ is stationary at the critical point. Vertical bars represent the statistical standard error (SE). **c** Experimental data showing the slope of the linear part of the Binder parameter [fitted between $30 < t < 400$ kicks and shown as dashed lines on **b**)] as a function of $K$. Error bars are the SE in the linear fitting. The critical point $K_c$ corresponds to a vanishing slope. In (**b**) and (**c**), the colors of the symbols match that of the corresponding circles in **a**.

green region in the phase diagram. This implies that the momentum distribution $\Pi_t(p)$ obeys a universal finite-time scaling law, allowing for studying the critical behavior of the Anderson transition. The scaling law reads

$$\Pi_t(p) = t^\alpha \mathcal{F}(p\, t^\beta, \delta_K\, t^\gamma), \qquad (3)$$

with $\delta_K = K - K_c$, $\mathcal{F}$ a universal function, and critical exponents $\alpha$, $\beta$, and $\gamma$. This universal scaling behavior holds for sufficiently long times (at short times, typically below $t_s = 30$ kicks in the present setup, the dynamics is not universal) and close enough to the critical point. It is the analog of the finite-size scaling in standard phase transitions, with time playing the role of the finite size that the system has been able to explore. The scaling form Eq. (3) can be interpreted as follows. After a short transient dynamics ($t < t_s$), and as long as $\delta_K\, t^\gamma$ is small enough, the momentum distribution at the critical point takes a universal shape $\Pi_t(p) \simeq t^\alpha \mathcal{F}(p\, t^\alpha, 0)$, see Fig. 1b, c). If $\delta_K = 0$, it stays critical. Otherwise, it will display this critical shape until $\delta_K t^\gamma$ is of order one, after which it will tend to a localized ($\delta_K < 0$) or diffusive ($\delta_K > 0$) shape.

The exponents in Eq. (3) are not independent but are related by the normalization of the momentum distribution and by the scaling of $p_{\text{loc}}$ and $D$[10]: in the localized phase ($\delta_K < 0$), $p_{\text{loc}} \propto |\delta_K|^{-\nu}$, defining the critical exponent $\nu$; in the diffusive phase ($\delta_K > 0$), with diffusion coefficient $D \propto \delta_K^s$, with critical exponent $s$. As shown in the MS, this implies $\alpha = \beta = -\nu/(s + 2\nu)$ and $\gamma = 1/(s + 2\nu)$. Moreover, the exponents $s$ and $\nu$ have been predicted to obey Wegner's scaling law $s = (d - 2)\nu$ for $d$-dimensional systems[19]. It has been verified in three-dimensional electronic systems[42], although with critical exponents in disagreement with the best numerical estimates[43] and state-of-the-art cold atoms experiments[14]. Studying the critical behavior Eq. (3) for the 4D Anderson transition allows us to experimentally determine the scaling exponents $\nu$ and $s$, and test Wegner's scaling law.

## Determination of the critical point

We experimentally studied the momentum distribution across the Anderson transition by following the path in the phase diagram indicated as a dashed black line in Fig. 2a. Our first goal is to locate the critical kick strength $K_c$; for this task, we take advantage of the above-mentioned scale invariance of the distribution shape at the critical point. It is convenient to introduce a Binder-like parameter that does not scale at the transition[44]. We observe that on the one hand, $\langle p^2\rangle_t = t^{-2\alpha}\Lambda(\delta_K\, t^\gamma)$, with $\Lambda(\tilde{\delta}) = \int d\bar{p}\, \bar{p}^2 \mathcal{F}(\bar{p}, \tilde{\delta})$ a universal scaling function, with $\tilde{\delta} = \delta_K t^\gamma$ and $\bar{p} = pt^\alpha$. On the other hand, we have

$\Pi_t(0)^2 = t^\alpha \mathcal{F}(0, \tilde{\delta})$. We thus define $B(t,K) = \Pi_t(0)^2\langle p^2\rangle_t = b(\delta_K\, t^\gamma)$ with $b(\tilde{\delta}) = \mathcal{F}(0, \tilde{\delta})^2\Lambda(\tilde{\delta})$ a universal scaling function. Being the product of the distribution's amplitude squared by its second moment, this Binder parameter is a constant if the shape of the distribution remains unchanged during its time evolution. The critical point $K_c$ can thus be identified as the value of $K$ at which the Binder parameter is constant, while for other values of $K$ it evolves towards the asymptotic localized (exponential) or diffusive (Gaussian) values. Importantly, using the Binder parameter allows us to locate the critical point without any prior assumption on the value of the critical exponents $\nu$ and $s$.

Figure 2b displays the Binder parameter calculated from the experimental data for three values of $K$. The curve in green circles is approximately horizontal for times $t > t_s = 30$, i.e. the Binder parameter is almost constant, signaling the proximity with the critical point. For smaller (respectively larger) values of $K$, corresponding to the blue (brown) circles, the Binder parameter is found to vary almost linearly with time, but with positive (resp. negative) slopes. Hence, we can use the slope of such curves as an indicator of the distance to the critical point. Adding more values of $K$ we obtain the plot displayed in the panel c, from which we can extract the value $K_c \simeq 5.62 \pm 0.15$, in good agreement with numerical simulations $K_c \simeq 5.45 \pm 0.05$ (error bars correspond to one standard deviation).

## Critical exponents

After finding the location of the critical point, we determine the critical exponents $\nu$ and $s$, see MS for details. Fig. 3a) shows the values of the kinetic energy $\langle p^2\rangle_t$, measured at various times as a function of $\delta_K$. Rescaling this data with trial values of the exponents allows us to optimize the collapse of $\langle p^2\rangle_t$ into a single curve, see Fig. 3b), using a least-square adjustment[45]. To assert the uncertainty of the critical exponents, we use a bootstrap method[46], which gives the joint probability distribution of the critical exponents shown in Fig. 3c). The optimal values of the exponents are $\nu_{\text{exp}} = 1.07 \pm 0.16$ and $s_{\text{exp}} = 2.22 \pm 0.38$ (error bars correspond to 95% confidence intervals), with Wegner's scaling law $s = (d - 2)\nu$ well satisfied for $d = 4$. This is in good agreement with the best numerical estimate $\nu_{\text{num}} = 1.156 \pm 0.014$ and $s_{\text{num}} = 2.312 \pm 0.028$ (using Wegner's scaling law) for a dimension four Anderson model[43], marked as a red square. The so-called self-consistent theory[47] is widely used to describe the localization physics in various settings, see ref. 48 for a review. While it gives a qualitative description of the Anderson physics in $d \leq 3$, it predicts an upper critical dimension equal to four, with $\nu_{\text{sc}} = 1/2$ and $s_{\text{sc}} = 1$ for any $d \geq 4$, marked as a red cross in Fig. 3c). This is in clear contradiction with

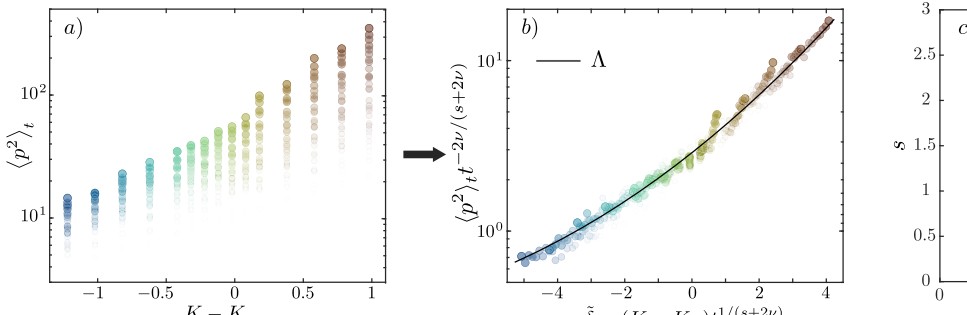

**Fig. 3 | Determination of the critical exponents $\nu$ and $s$. a** Kinetic energy $\langle p^2 \rangle_t$ as a function of the distance to the critical point $K - K_c$ at various times, with $K_c = 5.62$. Each value of $K - K_c$ is encoded in a different color, which goes from lighter to darker as time increases. Color code is similar to the one used in Fig. 2a), and reflects the distance to the critical point. **b** Same data after rescaling, the black line corresponds to the optimal scaling function $\Lambda(\tilde{\delta})$ using the optimal values $\nu_{\exp}$ and $s_{\exp}$, see MS. **c** Joint probability distribution of the exponents $\nu$ and $s$ obtained by bootstrap (see MS). We obtain $\nu_{\exp} = 1.07 \pm 0.16$ and $s_{\exp} = 2.22 \pm 0.38$ (blue dashed ellipse corresponds to the 95% confidence region, blue circle with error bars corresponding to the most probable value and 95% confidence intervals). The best numerical values for the critical exponents for the Anderson transition in $d = 4$ are $\nu_{\text{num}} = 1.156 \pm 0.014$[43] and $s_{\text{num}} = 2.312 \pm 0.028$ (using Wegner's scaling law), shown as a red square. The red cross represents the self-consistent theory prediction $\nu_{\text{sc}} = 1/2$ and $s_{\text{sc}} = 1$.

numerical simulations which indicate non-trivial $\nu > 1/2$ at least up to $d = 6$[43,49], due to disorder effects not included in this approximate approach. Our measurement is about 7 standard deviations away from the mean-field value $\nu = 1/2$, and thus constitutes the first experimental demonstration that $d = 4$ is not the Anderson transition's upper critical dimension.

## Two-parameter scaling law measurement

We now verify experimentally the two-parameter scaling law Eq. (3), which we find convenient to reexpress as

$$\Pi_t(p) = t^{-1/d} \mathcal{F}(p \, t^{-1/d}, \delta_K \, t^{1/\nu d}), \qquad (4)$$

where we use Wegner's scaling law $s = (d - 2)\nu$, with $d = 4$ and $\nu = 1.156$. We have shown in Fig. 1d) that, at criticality $K = K_c$, the momentum distribution is indeed scale-invariant. This property also holds for $K \neq K_c$ provided $\tilde{\delta} = \delta_K t^{1/d}$ is kept constant. This is shown in the top plots of Fig. 4a, b on the localized side of the transition $\tilde{\delta} < 0$ and in the top plots of Fig. 4c, d) on the diffusive side $\tilde{\delta} > 0$ (the corresponding values of $t$ and $\delta_K$ are given in Table 1—see MS). Having tested the two-parameter scale invariance, we can experimentally reconstruct the shape of the scaling function $\mathcal{F}(\tilde{p}, \tilde{\delta})$. While one-parameter scaling functions at criticality have been measured experimentally, e.g. for the 3D Ising[50,51] and 1D KPZ[52] universality classes, here we measured a two-parameter scaling function, which fully characterizes the dynamics close to the critical point. For that, we applied the scaling procedure to all the data displayed in Figs. 2 and 3; the result is shown in Fig. 4e). We observe that the data form a smooth surface corresponding to $\mathcal{F}(\tilde{p}, \tilde{\delta})$, evolving from a localized, exponential shape for $\tilde{\delta} < 0$ to a delocalized, Gaussian shape for $\tilde{\delta} > 0$.

## Discussion

In conclusion, we presented the first experimental study of a quantum phase transition in four dimensions, three of them being synthetic dimensions added to a physically 1D system. The critical scaling properties of the Anderson localization-delocalization transition were explicitly exhibited, including the full two-parameter scaling function, and the resulting critical exponents were found in good agreement with the numerical simulations of the "true" Anderson model (i.e. of disordered four-dimensional tight-binding lattices[43,49]), proving the universality of the Anderson transition in 4D. These results also confirm that $d = 4$ is not the upper critical dimension and may serve as a benchmark for the still missing quantitative theory of phase transitions belonging to the Anderson universality class. The techniques

developed here might be extended to higher dimensions and other kinds of models and physical systems, thus opening new ways to study exotic phases in high dimensions[53,54].

Finally, synthetic dimensions could allow for a first observation of multifractality, the properties of the system to exhibit complex, scale-dependent fluctuations that can be characterized by a spectrum of fractal dimensions rather than a single dimension. In the present setup, an experimentally accessible signature is the presence of a non-analytic behavior of $\mathcal{F}(\tilde{p}, 0) \propto a - b|\tilde{p}|^{d_f - 1}$ ($d_f \simeq 1.25$ in 3D and 0.9 in 4D[55]) at small $\tilde{p}$ and long time. This implies that a 4D system would have a much stronger signature of multifractality than a 3D one. It was shown in ref. 56 that in 3D hundreds of kicks were needed to observe this multifractal behavior, which can be attributed to the fact that the *physical system* is 1D, which significantly reduces the fluctuations responsible for multifractality (adding synthetic dimensions does not change this). Therefore, increasing the dimension of the physical underlying system could help make multifractality more observable. Constructing a *physically 2D* QpKR with incommensurable modulation frequencies to create synthetic dimensions, i.e. kick with two lasers along the $x$ and $y$ directions with modulated amplitudes, offers a promising alternative. While, to our knowledge, this configuration has not yet been studied in detail, it is reasonable to expect that it would significantly improve the conditions for observing multifractality experimentally.

## Methods

### Experimental section

**Experiment details and measurement procedure.** The experiments are performed with a thermal cloud of $\sim 10^6$ potassium atoms (isotope $^{41}$K) prepared by evaporative cooling in a crossed optical dipole trap at a temperature $T = 1.8 \, \mu\text{K}$. The cloud is kicked along the horizontal direction by a far-detuned optical standing wave. The final momentum distribution of the cloud, after the kick sequence, is measured using a "standard" time-of-flight (TOF) technique. In practice, this is done by allowing the kicked atomic cloud to expand freely for a given time, typically 20 ms. The spatial distribution of the atoms, captured through absorption imaging after the expansion, reflects the momentum distribution at the end of the kick sequence. Indeed, we checked that the size of the expanded cloud is systematically larger, by at least a factor seven than the initial one, at the beginning of the TOF. The average kinetic energy is determined by fitting the quantity $\Pi_t(p) \times p^2$ with a functional form (Lobkis-Weaver distribution[57,58]), following[21,58]. Experiments, corresponding to a given kick number and $(K, \varepsilon)$ couple, were typically repeated and averaged

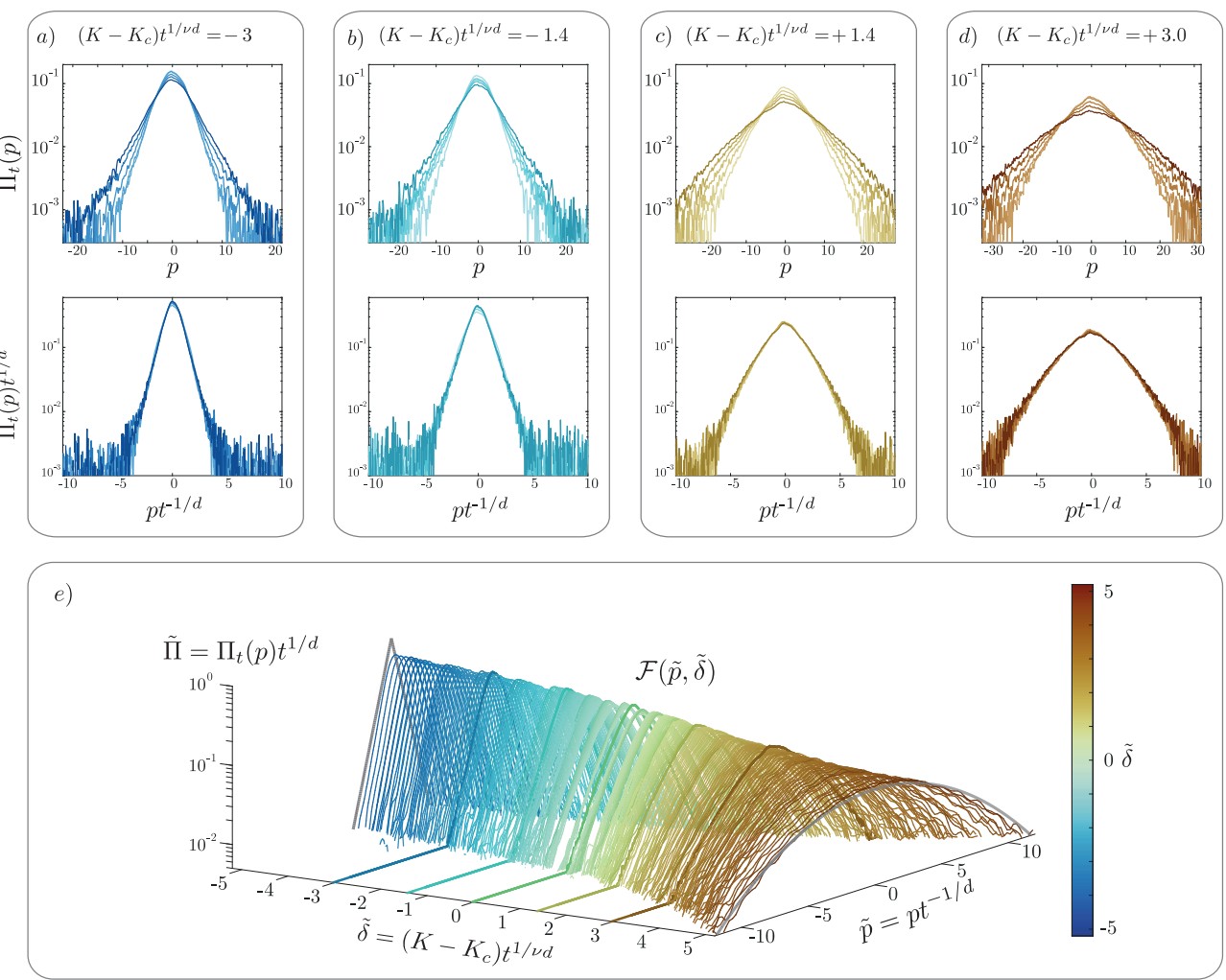

**Fig. 4 | Two-parameter scale invariance near criticality of the 4D Anderson transition.** Top panels: raw (top) and rescaled (bottom) momentum distributions, averaged over 20 experimental realizations, in the localized $\delta_K < 0$ [**a**, **b**)] and diffusive ($\delta_K > 0$) [**c**, **d**] regimes. In all panels, colors encode the value of the scaling parameter $\tilde{\delta}$, darker tones representing longer times. **e** Reconstruction of the two-parameter scaling function. Data obtained for various values of $K$ and $t$, properly rescaled, all fall on a smooth surface corresponding to the scaling function $\mathcal{F}(\tilde{p}, \tilde{\delta})$. The data of the bottom plot in (**a**, **b**, **c**, **d**), as well as the data at the critical point [Fig. 1d)], are shown as thick lines. The thick gray lines represent, respectively, the exponential (gaussian) distribution in the localized (delocalized) phase.

### Table 1 | Parameters for Fig. 4a–d

| $\bar{\delta}$ | $(\delta_K, t)$ | | | | |
|---|---|---|---|---|---|
| −3 | ( − 1.2, 69) | ( − 1.1, 103) | ( − 1, 161) | ( − 0.9, 262) | ( − 0.8, 451) |
| −1.4 | ( − 0.6, 50) | ( − 0.5, 117) | ( − 0.45, 190) | ( − 0.41, 293) | ( − 0.37, 470) |
| +1.4 | ( + 0.6, 50) | ( + 0.5, 117) | ( + 0.45, 190) | ( + 0.41, 293) | ( + 0.37, 470) |
| +3 | ( + 1.2, 69) | ( + 1.1, 103) | ( + 1, 161) | ( + 0.9, 262) | ( + 0.8, 451) |

This table resumes the experimental parameters used for testing the two-parameter scaling law in the vicinity of the critical point $K \simeq K_c$.

between three and six times. The relative dispersion of $\langle p^2 \rangle_t$ has a standard deviation of ~10% in the experiments, and is almost independent of the experimental parameters. Throughout the experiments, atomic densities are kept below $10^{12}$ cm$^{-3}$, corresponding to an interaction energy per particle of about 0.5 nK. This is much smaller compared to, e.g., the initial cloud temperature, and to the typical energies after the kick sequence (ranging typically between a few tens to several hundred recoil energies $E_{rec} \simeq 400$ nK). That makes interaction effects, which might alter the characteristics of the phase transition[59], negligible at the time scale of the kick sequence. To confirm that, we performed experiments with densities up to three times lower, without noticeable change in the measured momentum distributions.

**Pulsed optical standing wave.** The pulsed Kicked Rotor potential is obtained using a far-detuned 1D optical lattice generated with a pulsed laser system at 770.229 nm, with a detuning $\Delta = -61.2$ GHz with respect to the potassium D$_1$ line (770.108 nm). The laser light is obtained using a pulsed erbium-doped fiber amplifier at 1540.216 nm, which is frequency-doubled in free space using a single-passed Periodically Poled Lithium Niobate nonlinear crystal. The system produces optical pulses with a repetition frequency of 142.8 kHz (corresponding to $\hbar_e = 2.89$) and pulse durations of 20 ns (~0.3% of the pulse period) short enough that the motion of the atoms can be neglected during the pulse and the $\delta$−kicked potential approximation is well verified. Overall, the system delivers a maximum average optical power of 0.5 W, corresponding to peak laser powers up to ~170 W. The laser beam is passed

through an acousto-optic modulator (AOM) and delivered to the experiment with a polarization-maintaining (PM) optical fiber. The fiber output is collimated to a waist of 2.8 mm at the position of the atoms, and retro-reflected to create the pulsed optical standing wave.

**Stability of the kick strength.** Precise control of the kick strength and optical power stability is crucial for accurately measuring the critical exponents. The optical pulse amplitude modulations used to realize the synthetic dimensions are generated with an electronic arbitrary wave generator (AWG) which controls the RF power driving the AOM. To avoid optical and RF thermal effects, a mechanical shutter is placed before the AOM, and opened for a limited amount of time before the kick sequence. To achieve long-term stability, the AOM transfer function and the optical power after the fiber were regularly remeasured (typically every ten experimental cycles), using a beam pick-off on a fast photodiode, and the AWG modulation was recalculated accordingly. While the QpQKR experiments were performed with a thermal cloud and at low density, we used a $^{41}$K BEC in order to determine the value of the kick strength precisely. This was achieved by pulsing the lattice beams and observing an atom diffraction pattern in time-of-flight vs. the power of the lattice beams. We estimate the uncertainty of the measurement of the kick amplitude strength, as well as the achieved long-term stability, on the order of 1%. This value was inferred by analyzing the statistics of several hundreds of kick sequences, acquired with the photodiode, and confirmed by observing the long-term stability of the BEC diffraction patterns.

## Mapping on a disordered 4D model

Following refs. [29,30,35], the dynamics of the 4D-QpQKR can be mapped onto an effectively periodic one by introducing three extra dimensions, represented by canonically conjugated position and momentum operators $x_i$ and $p_i$, $i = 2, 3, 4$. In this respect, we define the effective Hamiltonian $\mathcal{H} = \frac{p^2}{2} + \omega_2 p_2 + \omega_3 p_3 + \omega_4 p_4 + K(1 + \varepsilon \cos(x_2) \cos(x_3) \cos(x_4)) \cos(x) \sum_n \delta(t - n)$, which corresponds to a *time-periodic* four-dimensional system, with linear kinetic energy for the extra dimensions. Furthermore, one can show[35] that the corresponding four-dimensional wavefunction $\Psi_t(x, x_2, x_3, x_4)$ with a "plane source" initial condition $\Psi_{t=0}(x, x_2, x_3, x_4) \equiv \psi_{t=0}(x)\delta(x_2)\delta(x_3)\delta(x_4)$ has the same dynamics along $x$ as that of the *physically one-dimensional* 4D-QpQKR with initial condition $\psi_{t=0}(x)$. Moreover, the Floquet eigenstates $\Phi$ of the one-period evolution operator associated with $\mathcal{H}$ with eigenvalue $e^{i\epsilon/\hbar_e}$ can be mapped onto eigenstates of a 4D tight binding model $\epsilon_{\mathbf{p}}\Phi_{\mathbf{p}} + \sum_{\mathbf{q}} t_{\mathbf{q}}\Phi_{\mathbf{p}-\mathbf{q}} = 0$, where $\mathbf{p} = (p, p_2, p_3, p_4)$ and $\mathbf{q}$ are the four-dimensional momenta on a hypercube. Here, the "on-site energies" are $\epsilon_{\mathbf{p}} = \tan\left[\frac{1}{2\hbar_e}\left(\epsilon - \frac{p^2}{2} - \omega_2 p_2 - \omega_3 p_3 - \omega_4 p_4\right)\right]$ and the short-range "hopping amplitudes" $t_{\mathbf{q}}$ are given by the four-fold Fourier transform of $\tan\left(\frac{K}{2\hbar_e}(1 + \varepsilon \cos(x_2) \cos(x_3) \cos(x_4)) \cos(x)\right)$. The hopping $t_{\mathbf{q}}$ is not limited just to nearest neighbors, but it is short-range and anisotropic, reduced by a factor $\varepsilon$ along the synthetic dimensions $p_i$ compared to the 'physical' one $p$. Furthermore, if $(\hbar_e, \omega_2, \omega_3, \omega_4, \pi)$ are incommensurate, $\epsilon_{\mathbf{p}}$ is a pseudo-random sequence, playing the role of a (correlated) disorder. The corresponding tight-binding model is therefore a (pseudo-)disordered Anderson model in momentum space.

While this model is not *strictly* identical to a bona fide Anderson model with uncorrelated local disorder and nearest neighbor hopping, it belongs to the same universality class and therefore presents the same critical physics at the Anderson transition[35]. Note that since the effective Hamiltonian is periodic in $x$ with period $2\pi$, the (physical) momentum $p$ can be split into a quasi-momentum $\beta\hbar_e$, $\beta \in [0.5, 0.5]$, and an "integer" part $n\hbar_e$, $n \in \mathbb{Z}$. The quasi-momentum $\beta$ is conserved during the dynamics and plays the role of a disorder realization in $\epsilon_{\mathbf{p}}$. Furthermore, the sum $\sum_{\mathbf{q}}$ above is only over the integer parts of the momentum components.

## Numerical determination of the phase diagram and $K_c$

The phase diagram and the location of the critical point are determined numerically by analyzing the variation of the kinetic energy's scaling function $\Lambda = \langle p^2 \rangle_t t^{-2/d}$ (see also ref. [60]). We performed numerical simulations of the QpQKR model and computed the time evolution of $\langle p^2 \rangle_t$ for different values of the $(K, \varepsilon)$ couple. The results were obtained by averaging the momentum distributions corresponding to $10^4$ values of the quasi-momentum $\beta$, uniformly distributed in $[-0.5, 0.5]$, and kick numbers up to 5000. The phase diagram was obtained by computing the slope of $\Lambda = \langle p^2 \rangle_t t^{-1/2}$ vs. $pt^{-1/4}$ (corresponding to $d = 4$). The critical region in Fig. 2 corresponds to the $(\varepsilon, K)$ couples for which $\Lambda$ is constant, yielding a zero-slope. The intersection between the critical line and the $(K, \varepsilon)$ path used in the experiment is obtained at $K = K_c = 5.45 \pm 0.05$.

## Scaling form and critical exponents

From Eq. (3), the normalization of the momentum distribution $\int dp \, \Pi_t(p) = 1$ for all $\delta_K$ and all times implies $\alpha = \beta$. Furthermore, this implies $\langle p^2 \rangle_t = t^{-2\alpha}\Lambda(\delta_K t^\gamma)$. On the one hand, in the localized phase ($\delta_K < 0$), $\langle p^2 \rangle_t$ will become time-independent at very long times, $\lim_{t\to\infty} \langle p^2 \rangle_t = p_{loc}^2$, defining the localization length $p_{loc}$. It diverges at the transition as $p_{loc} \propto |\delta_K|^{-\nu}$, defining the exponent $\nu$. This implies $\Lambda(\tilde{\delta}) \propto |\tilde{\delta}|^{2\alpha/\gamma}$ for $\tilde{\delta} < 0$ and $|\tilde{\delta}| \gg 1$, and thus $2\alpha/\gamma = -2\nu$. On the other hand, in the diffusive phase ($\delta_K > 0$) $\langle p^2 \rangle_t$ grows linearly in time at long times, $\lim_{t\to\infty} \langle p^2 \rangle_t = Dt$. The diffusion coefficient $D$ vanishes as $\delta_K^s$ close to the transition, defining the critical exponent $s$. This implies $\Lambda(\tilde{\delta}) \propto \tilde{\delta}^s$ for $\delta_K \gg 1$ and $s\gamma - 2\alpha = 1$. Finally, assuming Wegner's scaling law, which relates the exponents $s$ and $\nu$ as $s = (d - 2)\nu$, we would find $\alpha = -1/d$ and $\gamma = 1/\nu d$. Our experimental measurements of the critical exponents ($\nu_{exp} = 1.07 \pm 0.16$ and $s_{exp} = 2.22 \pm 0.38$, see text) are in good agreement with Wegner's law for $d = 4$.

## Critical exponents data analysis

**Fitting procedure for $\nu$ and $s$.** Our optimal scaling procedure uses the exponents $\nu$ and $s$ as fit parameters and consists in searching for the best collapse of the kinetic energy $\langle p^2 \rangle_t$ into a single scaling function $\Lambda$, for all values of $(K, \varepsilon)$ and $t$. We use a set of values of $\langle p^2 \rangle_t$ obtained along the $(K, \varepsilon)$ path shown in Fig. 2a), in the vicinity of $K_c$ (typically $|K - K_c| \leq 1$), and at times $t > 30$ kicks. We use trial values of the two critical exponents $\nu$ and $s$ to compute the two scaled quantities $t^{2\alpha}\langle p^2 \rangle_t$ and $\tilde{\delta} = \delta_K t^\gamma$, with $\alpha = -\nu/(s + 2\nu)$ and $\gamma = 1/(s + 2\nu)$, see text. We then compute an average curve $\Lambda_0(\tilde{\delta}; \nu, s)$, using a sliding average with a window of typically 1/20 of the total range of $\tilde{\delta}$. We interpolate this average curve and calculate the total variance of the quantity $\langle p^2 \rangle_t t^{2\alpha} - \Lambda_0(\tilde{\delta}; \nu, s)$. The result is shown in Fig. 5a), and the location of the minimum corresponds to the best-guess values of the exponents, $\nu_{fit}$ and $s_{fit}$. The corresponding horizontal and vertical slices are shown in Fig. 5b, c). Note that the shape of the reconstructed scaling function $\Lambda_0(\tilde{\delta}; \nu, s)$ does depend on the trial exponents $\nu$ and $s$. Our best estimate of the scaling function $\Lambda$ shown in Fig. 3b) is obtained from $\Lambda_0(\tilde{\delta}; \nu_{exp}, s_{exp})$ (with $\nu_{exp}$ and $s_{exp}$ obtained as described below).

**Bootstrap procedure.** To determine the optimal values of the critical exponents, $\nu_{exp}$ and $s_{exp}$, as well as the confidence intervals, we use a bootstrap method to take into account the impact of the statistical uncertainties of the experimental $\langle p^2 \rangle_t$ data, as well as that of the critical point location $K_c$[61]. Assuming a Gaussian probability distribution for these quantities, and given the standard deviations estimated above, we resample $\langle p^2 \rangle_t$ and $K_c$ and repeat the fitting procedure described in the previous paragraph to determine values of the $\nu$ and $s$ exponents. This procedure is repeated $10^4$ times, which provides the $(\nu, s)$ samples shown in Fig. 3c). We use these samples to determine the best estimates $\nu_{exp}$ and $s_{exp}$ of the critical exponents, as well as the

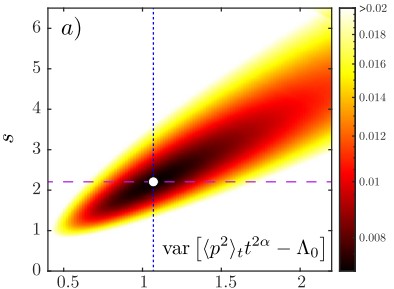
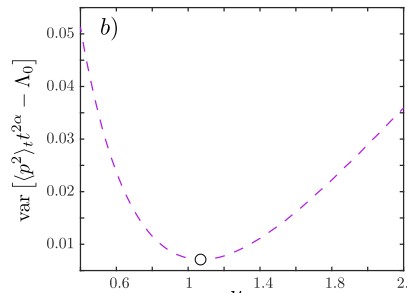
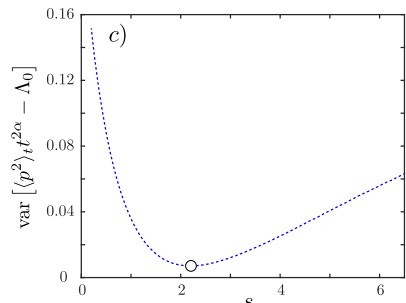

**Fig. 5 | Determination of the critical exponents. a** Color plot showing the variance of the dispersion of the scaled kinetic energy around $\Lambda_0$, as a function of the critical exponents $\nu$ and $s$. The minimum of this variance corresponds to the white point. The corresponding cross-sections are shown in (**b**) and (**c**).

corresponding confidence region (blue dashed ellipse) and the confidence intervals (shown as error bars). Due to the limited amount of experimental data, we estimate that our numerical resampling method provides realistic error estimations. Another option, standard bootstrapping (random sampling of the data with replacement), tends to underestimate confidence intervals in our case due to the small dataset (three to six averages per determination of $\langle p^2 \rangle_t$).

**Two-parameter scale invariance parameters**
Table 1 resumes the experimental values of the $(\delta_K, t)$ couples used for testing the two-parameter scale invariance, depicted in Fig. 4 a-d. The values were chosen such that $\bar{\delta} = \delta_K t^{1/\nu d}$ is kept constant along each line of the table.

## Data availability
The experimental data corresponding to the momentum distributions extracted from the time-of-flight measurements, as well as the processed data presented in all figures, are available in the ZENODO database under accession code https://doi.org/10.5281/zenodo.14278774.

## Code availability
Codes reproducing the results of numerical simulations, as well as additional information related to this paper, are available from the corresponding author, Radu Chicireanu, upon request.

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

## Acknowledgements

We thank C. Cherfan for his contributions to the early stages of the experiment, and V. Vuatelet for preliminary numerical simulations. We thank C. Hainaut, C. Chin, and X. Zhang for useful discussions, and A. Amo, C. Hainaut, and G. Lemarié for a careful reading of the manuscript. This work was supported by the Agence Nationale de la Recherche (ANR) through Research Grants MANYLOK No. ANR-18-CE30-0017 (P.S.) and Labex CEMPI (GrantNo. ANR-11-LABX-0007-01, P.S.), by CPER Wavetech (P.S.), and also by the PHC Cogito (project number 49149VE, A.R.) and CNRS IEA (A.R.) programs. The Contrat de Plan Etat-Region (CPER) WaveTech is supported by the Ministry of Higher Education and Research, the Hauts-de-France Regional Council, the Lille European Metropolis (MEL), the Institute of Physics of the French National Centre for Scientific Research (CNRS) and the European Regional Development Fund (ERDF).

## Author contributions

R.C. and A.R. designed the experiment. F.M. performed the experiments and data analysis. M.D. contributed to the setup construction and early experimental attempts. R.C. supervised the construction of the apparatus and the data collection effort. R.C., M.D., and F.M. performed numerical simulations. P.S. contributed to the realization of the SW laser and the experimental control sequence. A.R. performed theoretical work and developed scaling theory-based methodologies for data analysis. R.C. and A.R. wrote the first draft of the manuscript with inputs from J.-C.G. and all the authors contributed to the final draft.

## Competing interests

The authors declare no competing interests.
