## [Transparent Peer Review file · Nature Communications]

Observation of Quantum Criticality of a Four-Dimensional Phase Transition

Corresponding Author: Dr Radu Chicireanu

Version 0:

Reviewer comments:

Reviewer #1

(Remarks to the Author)

In this manuscript, Madani et al explore the critical behavior of Anderson localization in a synthetic 4D quasiperiodic quantum kicked rotor. While synthetic dimension has been widely employed to study physics in higher dimension ($d > 3$), this is, as claimed in the article, the first experimental study of localization phase transition in 4D. The experimental observations show clear agreement with the numerical simulations and the predicted critical scale invariance for $d = 4$. The measured critical exponents also allow testing the Wegner's relation in 4D. I see a path towards publication in Nature Communications. However, I would like the authors to address the following concerns before a definite recommendation being made.

1. As an extension of previous works on 3D (ref.14) and 2D (ref.34), it seems that the only difference is an additional frequency component introduced in the kick function $K(t)$. What new technique upgrades are required to study such 4D case, e.g., the condition of the atomic cloud, the repetition rate of the kick pulse, etc.? If the system is just conceptionally new, does it make sense to straightforwardly extend to effective 5, 6 or even 100 dimension? So, what's the limitation of the current setup? The authors might provide some clarification on this.
2. Some questions on experimental details: (i) While the procedure of the reconstruction of the momentum distribution with TOF imaging is very general in cold atom experiment, a clarification in supplement would make clearer sense to the readers. (ii) The uncertainty of the kick amplitude is claimed as on the order of 1%. What regime does the AOM work in with the laser power of 170W? Should it be affected by the thermal effect of AOM (i.e., diffraction efficiency is different after long-time run)? How do you estimate such an order of 1%? (iii) To make the experiment in weakly interacting regime, the atom density is on the order of 10^{12} cm^{-3} (is the unit correct?). Can the author provide a comparison of the interacting energy and the kick energy scale? This will help the readers to quantitatively understand the negligible role of interactions in current experiment.
3. The authors provide a brief description of the mapping from kicked rotor to a disordered 4D model in supplement. Without referring to ref.35, it is still hard to see straightforward disordered picture in momentum space. More details in supplement would be better, especially for the wide audiences/readers unfamiliar to the quasiperiodic kicked rotor system.
4. Since different driving frequencies are mapped into momentum dimensions to generate 4D disordered Hamiltonian, the universal critical behavior should be independent on the frequency choice. Have the authors checked the behavior with other different driving frequency sets? Or is the choice of driving frequencies case sensitive to the current experimental condition (atomic density, temperature, kick period, etc.)? It would be helpful to evidence the universality of the synthetic dimension method if the authors can provide such demonstration, even with numerical simulations.
5. In Fig.2, the authors use different spreading behaviors near the critical point to determine the 4D Anderson transition point. In localized regime ($4 < K < 5$), why does the slope of Binder parameter stay nearly as a constant? Does it mean that the spreading of the cloud shows nearly identical behaviors, or there are other different features for different K in this region?
6. In Figs.3 and 4, the authors compare the experimental observations to different theory predictions. The results are consistent with Wegner's scaling law, but roughly double of the values predicted by the self-consistent theory. What's the limitation of the self-consistent theory to describe high dimension disordered system? Some insights on this point would be good for readers to understand the full picture of such a system.

7. In the conclusion part, the authors claim that the results are 'in good agreement with the numerical simulations of the Anderson model (not that of the QpQKR)'. So Does it mean that the numerical simulations are not performed with the kicked rotor model? I'm confused about what model is used for such simulation.

Reviewer #2

(Remarks to the Author)

This work comprises the first experimental observation and characterization of the Anderson transition in four dimensions. The experimental system consists in a periodically kicked ultracold gas which undergoes a dynamical localization transition in momentum space. The kick intensity is specifically modulated using three incommensurate frequencies in order to obtain an effective Anderson model in $d=4$ dimensions. The theoretical and experimental foundations of the technique used are well understood by now, and some of the authors have a recognized experience employing it in the past to study experimentally Anderson localization in $d=2$ and the transition for $d=3$.

The reported experimental results demonstrate the predicted scale invariance around the vicinity of the transition for $d=4$, and provide experimental estimations for the localization length critical exponent ν , and for the exponent 's' of the diffusion coefficient. The former is found to be in agreement with numerical simulations, while the estimation of the latter allows the authors to show the validity of Wegner's relation between ν and 's' for $d=4$. Furthermore, the two-parameter scaling function that governs the transition is reconstructed from the experimentally measured momentum distribution of the gas.

As the authors correctly claim, this is the first experimental study of a quantum phase transition in four dimensions. In particular, the results experimentally confirm that $d=4$ is not the upper critical dimension for the Anderson transition, and spur further investigations of the Anderson transition in higher dimensions that may provide valuable insight to approach an accurate quantitative theory of the transition.

To my eyes, the study contains strong evidence ensuing from a sound analysis of the experimental data. Overall, I find the manuscript to be well written and presented, and the main message clearly conveyed to the readers. I do not hesitate to conclude that this work signifies a milestone in the field of localization-delocalization transitions, and also for the general study of quantum phase transitions.

I have nonetheless, a few minor comments, on which I elaborate below, that I would like the authors to consider.

(1) In page 3, at the end of the first paragraph, the authors write "...all eigenstates are of the same nature, either localized, critical, or diffusive, ...". I do not have an issue labelling the phases as localized, critical or diffusive, but the reference to 'diffusive eigenstates' I find rather unusual and potentially problematic. I would recommend the authors to simply use 'fully extended' or 'fully delocalized' eigenstates instead.

(2) In page 3, second paragraph, when referring to the phase diagram of Fig. 2(a), I think it should be clearly stated in the main text (and not only in the figure caption) that this is the numerically obtained phase diagram. Similarly, the caption of Fig. 2 should --clearly-- indicate that panels (b) and (c) follow from the analysis of experimental data. Since this figure mixes numerical simulation and experimental data is mandatory to be unambiguously clear to avoid any misunderstandings.

(3) In the caption of Fig. 2, in the second line ".. the slope of $\log(\dots)$...", they should indicate versus which variable this slope is measured.

(4) In relation to the linear fits of the Binder parameters versus t , I presume the fits are carried out between 30 and 400 kicks judging from Fig. 2(b), but I have failed to find the information on the final time used for the fits in the manuscript or the SM. I would suggest the authors to indicate that information explicitly, perhaps in the caption of Fig. 2.

(5) In page 4, at the end of first paragraph, "... the Binder parameter allows us to locate the critical point without any prior assumption on the critical scaling (namely, here, on the critical exponents...)". I would suggest to rewrite simply as "the Binder parameter allows us to locate the critical point without any prior assumption on the value of the critical exponents...". After all, a hypothesis for the scaling function around criticality needs to be and has been made.

(6) In page 4, the acronym UCD is used (standing for upper critical dimension), but it has not been defined in the manuscript.

(7) In Fig. 4(e), it would be advisable to explain in the caption what the thick gray lines in the plot represent (the exponential and Gaussian distributions, I presume).

(8) Finally, I have a question in relation to another important quantity that characterizes the critical point of the Anderson transition, namely the multifractal spectrum. Would it be possible to retrieve from the experimental data any information on the multifractal exponents? Such info would also be very valuable, insofar as the properties of the multifractal spectrum also bound the set of acceptable theories behind the transition.

Version 1:

Reviewer comments:

Reviewer #1

(Remarks to the Author)

The authors address all related concerns in detail in the response document, I recommend the publication in Nature Communications.

Reviewer #2

(Remarks to the Author)

I thank the authors for the reply to my comments and the changes implemented in the updated version of the manuscript. I consider that they have satisfactorily responded to all my queries and improved the manuscript accordingly.

I therefore recommend publication of this work in Nature Communications in its present form.

Reply to the Referee's comments

Farid Madani,¹ Maxime Denis,¹ Pascal Szriftgiser,¹ Jean Claude Garreau,¹ Adam Rançon,¹ and Radu Chicireanu¹

¹Univ. Lille, CNRS, UMR 8523 – PhLAM – Laboratoire de Physique des Lasers Atomes et Molécules, F-59000 Lille, France

(Dated: September 13, 2024)

We thank the Referees for their review, and for their appreciation of our work. We reply below to their comments in bold. The list of modifications is given at the end, and appear in red in the new version of the manuscript.

Reviewer 1

In this manuscript, Madani et al explore the critical behavior of Anderson localization in a synthetic 4D quasiperiodic quantum kicked rotor. While synthetic dimension has been widely employed to study physics in higher dimension ($d > 3$), this is, as claimed in the article, the first experimental study of localization phase transition in 4D. The experimental observations show clear agreement with the numerical simulations and the predicted critical scale invariance for $d = 4$. The measured critical exponents also allow testing the Wegner's relation in 4D. I see a path towards publication in Nature Communications. However, I would like the authors to address the following concerns before a definite recommendation being made.

We thank the Referee for their report, and for recognizing the novelty of our work on localization in high dimensions. We reply in details to their comments below.

1. As an extension of previous works on 3D (ref.14) and 2D (ref.34), it seems that the only difference is an additional frequency component introduced in the kick function $K(t)$. What new technique upgrades are required to study such 4D case, e.g., the condition of the atomic cloud, the repetition rate of the kick pulse, etc.? If the system is just conceptionally new, does it make sense to straightforwardly extend to effective 5, 6 or even 100 dimension? So, what's the limitation of the current setup? The authors might provide some clarification on this.

The difference with the previous studies is in the number of frequency components introduced in the kick function $\mathcal{K}(t)$. For unrelated reasons, since Ref. 34, we have completely modified our experimental setup (from a Cesium MOT to a Potassium ultracold atomic cloud/BEC). The main conceptual change from our previous studies is in the data analysis, which has allowed us to measure both critical exponents ν and s , which was not done in 3D in Ref. 14.

Note that the new experimental setup has nevertheless some important advantages compared to the old one, such as: (i) using lighter atoms allows for a higher kick frequency, and thus a shorter kick sequence duration – making the system less prone to decoherence (ii) in the 3D transition experiment, the kick number was limited to ~ 100 , by the fact that the atoms were falling outside the horizontal kick beam. Here, the higher kick frequency allows to explore higher kick numbers, which is essential for investigating the slower dynamics in 4D (iii) the new *pulsed* laser we developed also has higher peak power, which allows us to work with shorter pulse duration (by about two orders of magnitude compared to the 3D experiments), making the atomic movement completely negligible during the kicks (iv) the atomic cloud is smaller in size, which, together with a larger beam waist, reduces inhomogeneity effects.

Concerning the possibility of extending to even higher dimensions. This is indeed possible in principle. There are however two related practical difficulties. First, the dynamics becomes slower and slower as d increases, since, for instance, the kinetic energy at the critical point scales as $t^{2/d}$. This implies that one needs to go to longer times to see some observable effects. Related to this, the broadening of the critical momentum distribution is slower, meaning that the finite width of the initial momentum distribution could hide the critical dynamics for longer times (compared to smaller dimensions where the dynamics is faster). This would require preparing colder initial samples, yet sufficiently dilute to keep interactions negligible.

2. Some questions on experimental details: (i) While the procedure of the reconstruction of the momentum distribution with TOF imaging is very general in cold atom experiment, a clarification in supplement would make clearer sense to the readers. (ii) The uncertainty of the kick amplitude is claimed as on the order of 1%. What regime does the AOM work in with the laser power of 170W? Should it be affected by the thermal effect of AOM (i.e., diffraction efficiency is different after long-time run)? How do you estimate such an order of 1%? (iii) To make the experiment in weakly interacting regime, the atom density is on the order of 10^{12}cm^{-2} (is the unit correct?). Can the author provide a comparison of the interacting energy and the kick energy scale? This will help the readers to quantitatively understand the negligible role of interactions in current experiment.

To address the reviewer’s questions, as well as the editor’s suggestions, about the details of our experiment, we reorganized the experimental sections of the supplementary materials (SM) and added the information requested. The new version contains three new sections (“Experiment details and measurement procedure”, “Pulsed optical standing wave” and “Stability of the kick strength”) which are replacing two of the old ones (“Experiment details” and “Measurement procedure and experimental uncertainties.”).

We summarize here the answers to the three specific points raised by the reviewer which were addressed in this new version:

(i) We added a short description of the time-of-flight procedure in the SM (“Experiment details”).

(ii) We point out that we use a ‘natively’ pulsed laser system, with pulse duration of 20 ns and peak pulse powers up to 170 W (see SM), which corresponds to an average optical power below 0.5 W. In addition, a mechanical shutter is used before the AOM. The shutter lets the light reach the AOM only for the limited amount of time when the kick sequence is performed – typically of a few tens of ms for each experimental cycle (30 s). That makes the AOM thermal effects due to the incoming light power negligible. The short-term stability of the pulses is measured using a fast photodiode placed before the experimental chamber, and is well below 1%. Long term drifts can be more problematic, because of the thermal stability of the whole laser system (not just the AOM). We correct slow drifts of the power by regularly measuring and correcting it (typically every ten experimental runs, as mentioned in the SM). The overall stability is inferred by analysing the statistics of several hundreds of kick sequences, acquired with the photodiode, and it was ultimately confirmed by observing the long-term stability of the BEC diffraction patterns. To clarify these points, we added more details in the SM (“Pulsed optical standing wave” and “Stability of the kick strength”).

(iii) The peak atomic density is on the order of 10^{12} cm^{-3} (we corrected the unit), which gives an interaction energy per particle of about 0.5 nK. This is much smaller compared to, e.g., the initial cloud temperature (1 μK), and to the typical energies after the kick sequence (ranging typically between a few tens to several hundreds recoil energies $E_{\text{rec}} \simeq 400 \text{ nK}$). These values were added in the new SM version (“Experiment details”).

3. *The authors provide a brief description of the mapping from kicked rotor to a disordered 4D model in supplement. Without referring to ref.35, it is still hard to see straightforward disordered picture in momentum space. More details in supplement would be better, especially for the wide audiences/readers unfamiliar to the quasiperiodic kicked rotor system.*

We have modified this discussion in the supplemental materials, and explained better in what sense the Floquet eigenstates are eigenstates of a disordered systems.

4. *Since different driving frequencies are mapped into momentum dimensions to generate 4D disordered Hamiltonian, the universal critical behavior should be independent on the frequency choice. Have the authors checked the behavior with other different driving frequency sets? Or is the choice of driving frequencies case sensitive to the current experimental condition (atomic density, temperature, kick period, etc.)? It would be helpful to evidence the universality of the synthetic dimension method if the authors can provide such demonstration, even with numerical simulations.*

The universal behavior is indeed independent of the frequency choices (as long as they obey the necessary incommensurability conditions discussed in the text). This has been checked thoroughly both experimentally and numerically in the case of the 3D QpQKR in EPL 87, 37007 (2009) and PRL 108, 095701 (2012). We have added a short note about this point, and the corresponding references, in the text.

5. *In Fig.2, the authors use different spreading behaviors near the critical point to determine the 4D Anderson transition point. In localized regime ($4 < K < 5$), why does the slope of Binder parameter stay nearly as a constant? Does it mean that the spreading of the cloud shows nearly identical behaviors, or there are other different features for different K in this region?*

The Binder parameter is expected to follow a linear trend only in the vicinity of the critical point, and only for a limited time – except right at $K = K_c$. This is well-reflected by the size of the fitting error bars shown in Fig.2.c. Away from the phase transition and at ‘long times’, the Binder parameter will eventually tend towards a constant asymptotic value, characterized by the shape of the localized and diffusive distributions. Thus the slope of its variation will decrease with the number of kicks – making the linear fit procedure less and less justified.

Furthermore, in practice we can expect certain effects to lead to a faster decrease of the Binder slope on the localized side of the transition, and thus to a certain asymmetry between the localized

and diffusive regimes, visible in Fig.2.c. Indeed, in the localized regime, the localization time gets shorter and shorter as K decreases, to the point that it becomes on the order, or smaller than the ‘non-universal’ time scale t_s . Moreover, the localization length also decreases, to the point where it approaches the width of the initial (thermal) momentum distribution. Both effects tend to decrease the ‘apparent’ slope of the variation of the Binder parameter in the $K < 5$ range. We thus believe that no relevant physics, related to the critical behavior, occurs in that range.

6. In Figs.3 and 4, the authors compare the experimental observations to different theory predictions. The results are consistent with Wegner’s scaling law, but roughly double of the values predicted by the self-consistent theory. What’s the limitation of the self-consistent theory to describe high dimension disordered system? Some insights on this point would be good for readers to understand the full picture of such a system.

The self-consistent theory of localization gives an approximate (self-consistent) equation for the diffusion coefficient. It captures qualitatively the Anderson transition in dimensions $d = 3$, and the absence of a transition in dimensions $d = 1$ and $d = 2$. For dimensions $d \geq 4$, it predicts that the critical exponent $\nu = 1/2$ does not change, which is known to be false. This wrong prediction comes necessarily from the effects of disorder that are not included in the self-consistent equation, but we are not aware of specific studies on that particular point in high dimensions. We have expanded our discussion of the self-consistent theory in the text.

7. In the conclusion part, the authors claim that the results are ‘in good agreement with the numerical simulations of the Anderson model (not that of the QpQKR)’. So Does it mean that the numerical simulations are not performed with the kicked rotor model? I’m confused about what model is used for such simulation.

Our phrasing was indeed unclear. What we meant is that the critical exponents we have measured are in good agreement with those obtained in Refs 44 and 49 (numbering of the new version of the manuscript) from simulations of disordered four-dimensional tight-binding lattices (and not simulations of a 4D QpQKR). This is of course expected from the universality of the transition. This has been clarified in the text.

Reviewer 2

This work comprises the first experimental observation and characterization of the Anderson transition in four dimensions. The experimental system consists in a periodically kicked ultracold gas which undergoes a dynamical localization transition in momentum space. The kick intensity is specifically modulated using three incommensurate frequencies in order to obtain an effective Anderson model in $d=4$ dimensions. The theoretical and experimental foundations of the technique used are well understood by now, and some of the authors have a recognized experience employing it in the past to study experimentally Anderson localization in $d=2$ and the transition for $d=3$.

The reported experimental results demonstrate the predicted scale invariance around the vicinity of the transition for $d=4$, and provide experimental estimations for the localization length critical exponent ν , and for the exponent ‘s’ of the diffusion coefficient. The former is found to be in agreement with numerical simulations, while the estimation of the latter allows the authors to show the validity of Wegner’s relation between ν and ‘s’ for $d=4$. Furthermore, the two-parameter scaling function that governs the transition is reconstructed from the experimentally measured momentum distribution of the gas.

As the authors correctly claim, this is the first experimental study of a quantum phase transition in four dimensions. In particular, the results experimentally confirm that $d=4$ is not the upper critical dimension for the Anderson transition, and spur further investigations of the Anderson transition in higher dimensions that may provide valuable insight to approach an accurate quantitative theory of the transition.

To my eyes, the study contains strong evidence ensuing from a sound analysis of the experimental data. Overall, I find the manuscript to be well written and presented, and the main message clearly conveyed to the readers. I do not hesitate to conclude that this work signifies a milestone in the field of localization-delocalization transitions, and also for the general study of quantum phase transitions.

We thank the Referee for their very positive evaluation of our work and we are grateful for the recognition of its significance as a milestone in the study of localization-delocalization transitions and quantum phase transitions.

I have nonetheless, a few minor comments, on which I elaborate below, that I would like the authors to consider.

(1) In page 3, at the end of the first paragraph, the authors write "...all eigenstates are of the same nature, either localized, critical, or diffusive, ...". I do not have an issue labelling the phases as localized, critical or diffusive, but the reference to 'diffusive eigenstates' I find rather unusual and potentially problematic. I would recommend the authors to simply use 'fully extended' or 'fully delocalized' eigenstates instead.

We agree with the Referee that the terminology "diffusive eigenstate" is misleading, and we have corrected it in the text.

(2) In page 3, second paragraph, when referring to the phase diagram of Fig. 2(a), I think it should be clearly stated in the main text (and not only in the figure caption) that this is the numerically obtained phase diagram. Similarly, the caption of Fig. 2 should –clearly– indicate that panels (b) and (c) follow from the analysis of experimental data. Since this figure mixes numerical simulation and experimental data is mandatory to be unambiguously clear to avoid any misunderstandings.

We agree with the Referee and have modified the text and captions accordingly.

(3) In the caption of Fig. 2, in the second line "... the slope of log(...)", they should indicate versus which variable this slope is measured.

We agree with the Referee and have modified the text and captions accordingly.

(4) In relation to the linear fits of the Binder parameters versus t , I presume the fits are carried out between 30 and 400 kicks judging from Fig. 2(b), but I have failed to find the information on the final time used for the fits in the manuscript or the SM. I would suggest the authors to indicate that information explicitly, perhaps in the caption of Fig. 2.

The Referee's assumption is correct, and we have given the fit range in the caption.

(5) In page 4, at the end of first paragraph, "... the Binder parameter allows us to locate the critical point without any prior assumption on the critical scaling (namely, here, on the critical exponents...)." I would suggest to rewrite simply as "the Binder parameter allows us to locate the critical point without any prior assumption on the value of the critical exponents...". After all, a hypothesis for the scaling function around criticality needs to be and has been made.

We agree and have modified accordingly.

(6) In page 4, the acronym UCD is used (standing for upper critical dimension), but it has not been defined in the manuscript.

We have written UCD in full, since this acronym is not used anywhere else in the manuscript.

(7) In Fig. 4(e), it would be advisable to explain in the caption what the thick gray lines in the plot represent (the exponential and Gaussian distributions, I presume).

This is now corrected in the caption.

(8) Finally, I have a question in relation to another important quantity that characterizes the critical point of the Anderson transition, namely the multifractal spectrum. Would it be possible to retrieve from the experimental data any information on the multifractal exponents? Such info would also be very valuable, insofar as the properties of the multifractal spectrum also bound the set of acceptable theories behind the transition.

It has been proposed by D. Delande and collaborators (Phys. Rev. A 100, 043612 (2019)) that at least one of the multifractal exponents (namely, the exponent d_2) could in principle be measured in a QpQKR experiment by carefully measuring the peak at zero momentum of the momentum distribution. In particular, given the value of the multifract exponent, it might even be easier to observe this peak in 4D than in 3D.

However, it was shown in that study that, at least in the 3D QpQKR, one needs a number of kicks that appear to be undoable experimentally (hundreds of thousands). This is because in this setup, one only observes the physics in one dimension, which reduces the amplitudes of the fluctuations. It might be possible to alleviate this difficulty by using a physically 2D QpQKR in addition to synthetic dimensions (i.e. kick along two physical directions x and y and modulate the kicks amplitudes).

We have added this discussion in the conclusion.

List of corrections (in red in the new version of the manuscript):

- p. 2: added note on the independence of universality on the frequencies (point 4 of Referee 1)
- p. 3: changed “diffusive eigenstates” to “delocalized eigenstates” (point 1 of Referee 2)
- p. 3: precisions about the phase diagram in the main text, and modification of caption of Fig. 2 (points 2, 3 and 4 of Referee 2)
- p. 4: modification of the sentence about the critical scaling (point 5 of Referee 2)
- p. 4: changed “UCD” to “upper critical dimension” (point 6 of Referee 2) and the discussion of the self-consistent theory (point 5 of Referee 1)
- p. 5: added description of the gray lines in the caption of Fig. 4 (point 7 of Referee 2)
- p. 6: modified sentence about simulations in the conclusion (point 7 of Referee 1)
- p. 6: added one paragraph on perspectives about multifractality after the conclusion (point 8 of Referee 2)
- p. 9: added more experimental details (point 2 of Referee 1) and reorganized the corresponding sections of the SM
- p. 9: added more details on the mapping to disordered systems (point 3 of Referee 1)